# Association between hypoglycemic agent use and the risk of occurrence of nonalcoholic fatty liver disease in patients with type 2 diabetes mellitus

**Choungwon Jung**[1], **Soyoung Park**[1], **Hyunah Kim**[1,2]*

**1** College of Pharmacy, Sookmyung Women's University, Seoul, Republic of Korea, **2** Drug Information Research Institute, Sookmyung Women's University, Seoul, Republic of Korea

* hyunah@sookmyung.ac.kr

**Data Availability Statement:** The data underlying the results presented in the study are available from the National Health Insurance Service-National Sample Cohort (NHIS-NSC) 2.0 database

## Abstract

Nonalcoholic fatty liver disease (NAFLD) is a growing health concern with increasing prevalence and associated health impacts. Although no approved drugs are available for the NAFLD treatment, several hypoglycemic agents have been investigated as promising therapeutic agents. We aimed to compare the risk of occurrence of NAFLD with respect to the use of different hypoglycemic agents in patients with type 2 diabetes. This retrospective cohort study used data from the National Health Insurance Service–National Sample Cohort of South Korea. Participants newly diagnosed with type 2 diabetes (2003–2019) were included in this study. Two new user-active comparator cohorts were assembled: Cohort 1, new users of thiazolidinediones (TZD) and dipeptidyl peptidase-4 inhibitors (DPP-4i), and Cohort 2, new users of sodium-glucose cotransporter-2 inhibitors (SGLT-2i) and DPP-4i. The occurrence of NAFLD was defined based claims that include diagnostic codes. Hazard ratios (HRs) and 95% confidence intervals (CIs) were estimated using Cox proportional hazard models in 1:3 propensity score (PS)-matched cohorts. For 65,224 patients newly diagnosed with type 2 diabetes, the overall prevalence of NAFLD was 42.6%. The PS-matched Cohort 1 included 6,351 and 2,117 new users of DPP-4i and TZD, respectively. Compared to DPP-4i, TZD use was associated with the decreased risk of NAFLD (HR, 0.66; 95% CI: 0.55–0.78). Cohort 2 consisted of 6,783 and 2,261 new users of DPP-4i and SGLT-2i, respectively; SGLT-2i use was associated with a decreased risk of NAFLD (HR, 0.93; 95% CI: 0.80–1.08). This population-based cohort study supports the clinical implications of prioritizing TZD and SGLT-2i over DPP-4i in reducing the risk of occurrence of NAFLD in patients with type 2 diabetes. However, the findings lacked statistical significance, highlighting the need for further verification studies.

## Introduction

Non-alcoholic fatty liver disease (NAFLD)—a condition in which excessive fat accumulates in the liver in the absence of alternative causes of steatosis—is a growing public health concern

in Republic of Korea. https://nhiss.nhis.or.kr/bd/ab/bdaba000eng.do.

**Funding:** This work was supported by the National Research Foundation of Korea (NRF) grant funded by the Korean government (MSIT) [grant number NRF-2021R1F1A1052976].the funder of this study, National Research Foundation of Korea, had no role in study design, data collection and analysis, decision to publish, or preparation of the manuscript.

**Competing interests:** The authors have declared that no competing interests exist.

due to its increasing prevalence and associated health impacts [1]. NAFLD encompasses various conditions within its progression: simple steatosis, nonalcoholic steatohepatitis (NASH), liver fibrosis, and cirrhosis. The global prevalence of NAFLD is approximately 25% and is estimated to increase by 33.5% in the next 10 years [2,3]. In South Korea, the number of patients with NAFLD increased 43% from approximately 280,000 in 2017 to 400,000 in 2021 [4].

Numerous studies have demonstrated that NAFLD is not only associated with higher risk of liver-related morbidity and mortality, but also with the development of extra-hepatic complications [5]. Therefore, preventing the development and progression of NAFLD is crucial, especially in patients with type 2 diabetes, who have significantly higher risk of NAFLD compared to those without type 2 diabetes [6,7].

As no pharmacological agents have been approved for NAFLD currently, lifestyle modifications, such as weight loss and exercise, remain as the cornerstone of its management. Meanwhile, several hypoglycemic agents have been studied as potential options for NAFLD treatment and prevention [8]. Thiazolidinediones (TZD), pioglitazone in particular, have been shown to have beneficial effects on liver function and may reduce the risk of NAFLD by enhancing insulin sensitivity and ameliorating liver fat accumulation [9]. Additionally, growing evidence indicates that sodium glucose cotransporter-2 inhibitors (SGLT-2i) significantly decrease liver fat and improve liver function in patients with NAFLD [10,11].

Previous studies have mainly investigated the effect of hypoglycemic agents on ameliorating NAFLD. In contrast, potential effects of the drugs on patients without pre-existing NAFLD have been poorly understood. Several studies have examined whether TZD or SGLT-2i have a beneficial effect on preventing NAFLD and related outcomes, but their findings have been inconsistent [12,13]. Therefore, we aimed this study assess the risk of NAFLD occurrence with respect to the use of TZD and SGLT-2i, compared to dipeptidyl peptidase-4 inhibitors (DPP-4i) in patients with type 2 diabetes. We expect this study to support the development of effective pharmacotherapy strategies in patients with type 2 diabetes by providing further evidence to support the role of TZD and SGLT-2i in managing the risk of NAFLD occurrence.

## Materials and methods

### Data source

This study utilized data from the National Health Insurance Service-National Sample Cohort (NHIS-NSC) 2.0 database. The NHIS-NSC is an administrative health dataset established by the NHIS—a compulsory national health insurance service that provides coverage for all citizens in South Korea. NHIS-NSC 2.0 database includes the pseudonymized health data of approximately one million health insurance beneficiaries collected from 2002 to 2019, which comprises four sub-databases: insurance eligibility, medical treatments, medical care institutions, and general health examinations.

### Study design

We designed a retrospective cohort study to examine the association between the use of hypoglycemic agents and the risk of NAFLD. The active comparator, new-user study design was applied to minimize the possible bias arising from the non-experimental study design. New-users of DPP-4i were set as an active comparator, since DPP-4i has been known to have a neutral effect on body weight and liver fat, and also because it is considered a therapeutic alternative to TZD or SGLT-2i as a second-line drug for type 2 diabetes [14]. Cohort 1 included new users of DPP-4i or TZD from 2007 to 2019 and Cohort 2 included new users of DPP-4i or SGLT-2i from 2013 to 2019. The study period for each cohort was established based on the earliest approval date of the study drugs (i.e., DPP-4i and TZD in Cohort 1, and DPP-4i and

**Table 1. Comparator and treatment group and study period of each cohort.**

| Cohort | Comparator *vs.* treatment | Study period |
| --- | --- | --- |
| 1 | DPP-4i *vs.* TZD | 2007.09.01~2019.12.31 |
| 2 | DPP-4i *vs.* SGLT-2i | 2013.11.01~2019.12.31 |

Abbreviations: DPP-4i, dipeptidyl peptidase-4 inhibitors; TZD, thiazolidinediones; SGLT-2i, sodium glucose cotransporter-2 inhibitors.

SGLT-2i in Cohort 2) in South Korea (Table 1). This study granted a waiver for the need of informed consent from individuals by the Institutional Review Board of Sookmyung Women's University, Seoul, Republic of Korea, since the NHIS masked personal identifiers to de-identified join keys to secure ethical clearance (No. SMWU-2204-HR-016).

## Study population

Patients newly diagnosed with type 2 diabetes mellitus between 2003 and 2019 were included in this study. Presence of type 2 diabetes was defined as those who had both claims with diagnostic codes of type 2 diabetes and a history of hypoglycemic agent use. Diagnosis codes used to identify patient with type 2 diabetes were E11, E12, E13, and E14 based on the 10th revision of the International Classification of Diseases (ICD-10).

Among patients with newly diagnosed with type 2 diabetes, those who had been prescribed at least once with one of three classes of hypoglycemic agents (i.e., DPP-4i, TZD, and SGLT-2i) were included as the eligible patients. Eligible patients with newly diagnosed type 2 diabetes were then included in the corresponding cohort based on the comparator and treatment drugs. Within each cohort, patients were assigned to the comparator or treatment groups based on the initial hypoglycemic agents used during the study period.

The index date for each patient was defined as the date on which the study drug was first prescribed. The following patients were excluded from the study: (i) those who already had type 2 diabetes in 2002; (ii) those that were prescribed the study drug within a year before the index date (iii) those aged <20 years; (iv) those that were prescribed comparator and treatment drugs simultaneously at the index date; (v) patients with prescription durations of <30 days after the index date to ensure the inclusion of only adherent users; and (vi) individuals with a history of chronic liver diseases including NAFLD within the previous year of the index date. Diagnostic codes based on the ICD-10 used to define the history of NAFLD and chronic liver diseases are listed in S1 Table.

## Exposure

To define new users of each hypoglycemic agent, prescription information was extracted based on the drug codes of ingredients for each class of hypoglycemic agents, which were available in Korea from 2003 to 2019: pioglitazone, rosiglitazone, and lobeglitazone for TZD; sitagliptin, vildagliptin, alogliptin, gemigliptin, teneligliptin, evogliptin, anagliptin, linagliptin, and saxagliptin for DPP-4i; dapagliflozin, ipragliflozin, empagliflozin, and ertugliflozin for SGLT-2i.

Drug exposure was assessed based on as-treated approach to consider only the person-time when patients are on the initially assigned treatment is evaluated. Accordingly, patients were followed up from the index date and were censored at the earliest date among when deviation from the initial exposure happened (discontinuation of the initial drug or addition of the other study drug), end of study date, or death.

For fixed-dose combination hypoglycemic agents, each component was considered respectively as individual's drug exposure. Accordingly, patients in Cohort 1 who used fix-dosed

combination drugs consisting of DPP-4i and TZD were excluded since they were considered as those prescribed with comparator and treatment drugs at the same time. It is of note that fixed-dosed combination drugs of DPP-4i and SGLT-2i were not reimbursed in South Korea as of the end of the study period. Continuous use of the drug was defined as a case where the subsequent prescription starts within the grace period, which was set as the 1.5 times of duration of previous prescription, after the end of the previous prescription.

## Outcomes

The outcome of interest was NAFLD, which was defined based on claims that included ICD-10 codes of K76.0 (NAFLD) or K75.8 (NASH). If not specified with diagnostic codes, "NAFLD" in the following text refers to the outcome that encompasses both NAFLD (K76.0) and NASH (K75.8). The date of outcome occurrence was defined as the start date of the claim that had diagnosis codes indicating NAFLD. Individuals were followed up from the index date to the earliest date among the occurrence of NAFLD, discontinuation of the study drug, the date of death, or the end of the study period. Graphical depiction of the study design is shown in S1 Fig.

## Confounding variables

In the propensity score matching process, we controlled for potential factors that could impact the use of specific hypoglycemic drugs as well as the risk of NAFLD. Confounding variables included sex, age, comorbidities, concomitant medications, the number of different classes of hypoglycemic agents taken, and surrogate variables for severity of diabetes—duration of diabetes, presence of diabetic complications. Covariate assessment window was 180 days prior to the index date.

As comorbidities, the presence of dyslipidemia, hypertension, sleep apnea, hypothyroidism, polycystic ovary syndrome, myocardial infarction, ischemic heart disease, or chronic kidney disease were considered. The Charlson Comorbidity Index (CCI) was included as a covariate as well. Concomitant medications included hypoglycemic agents other than study drugs (such as metformin, sulfonylureas, alpha-glucosidase inhibitors, meglitinides, and insulin) and medications known to affect steatosis in the liver (such as methotrexate, tamoxifen, glucocorticoids, valproic acid, amiodarone, and hormone replacement therapy). Concomitant medications were defined as medications that individuals take for at least one overlapping day within the follow-up period.

BMI of the index date was also included in the propensity score matching. Baseline BMI of each patient was defined as a single-point value of BMI measured on the closest date from the index date within the 2 years of time window: a period between 1 year prior and after the index date. This definition of baseline BMI was to incorporate as much available data as possible, given that BMI records were only available on an annual or biennial basis in NHIS-NSC database [15,16]. Observations with missing values in the BMI data were excluded from the analysis.

## Statistical analysis

Propensity score matching was performed within each cohort to minimize potential selection bias. The propensity score of each individual was calculated by considering covariates through a multivariate logistic regression analysis. Propensity score matching was performed using nearest neighbor matching at a 1:3 ratio to secure statistical power and minimize the number of dropped samples. The balance between the comparator and treatment groups was evaluated

using the standardized mean difference, and a value of < 0.1 was considered indicative of well-balanced propensity scores [17].

Patient baseline characteristics are presented as means and standard deviations for continuous variables and as frequencies and percentages for categorical variables. A Cox proportional hazards model was used to obtain the hazard ratio (HR) and 95% confidence interval (CI) for NAFLD occurrence. We performed a subgroup analysis based on patients' age, sex, BMI, year of index date, and ingredients of the hypoglycemic drugs.

To evaluate the potential impact of a delayed NAFLD diagnosis following the cessation of drug exposure, we performed a sensitivity analysis with latency periods of 30, 60, 90, and 120 days. All statistical analyses were performed using SAS Enterprise Guide 7.1 (SAS Institute Inc., Cary, NC, USA).

## Results

Approximately one million participants were included in the 2002–2019 NHIS-NSC sample cohort database, and 117,727 of the population had type 2 diabetes. A total of 105,061 patients were newly diagnosed with type 2 diabetes patients between 2003 and 2019. After excluding 39,837 participants without any prescriptions of study drugs (i.e., DPP-4i and TZD, and SGLT-2i), 65,224 individuals were finally included as the eligible patients. A total of 65,224 patients were grouped into each group, in a non-mutually exclusive manner, of 50,094, 19,571, and 11,127 patients with a history of using DPP-4i, TZD, and SGLT-2i, respectively. Subsequentially, two groups of patients with corresponding comparator and control drugs were grouped as the sum of sets to construct Cohort 1 and Cohort 2 (e.g., Cohort 1 included those prescribed with DPP-4i or TZD at least once in 2003–2019) (Fig 1).

The overall prevalence of NAFLD among the study participants (n = 65,224) was 42.6%; it was higher in females than in males and higher in those aged 19–44 years compared to the other age groups. NAFLD was more prevalent in groups with higher BMI (35.8%, 42.1%, 47.6%, and 54.5% in groups with BMI<23 kg/m$^2$, 23≤BMI<25 kg/m$^2$, 25≤BMI<30 kg/m$^2$, and BMI≥30 kg/m$^2$ group, respectively, $p$<0.001) (S2 Table).

After further application of the exclusion criteria within each cohort, the number of eligible patients was 24,098 in Cohort 1 and 20,188 in Cohort 2 (Figs 2 and 3). Prior to propensity score matching in Cohort 1, the average age of DPP-4i new users and new TZD users was similar at approximately 58 years. All covariates were well-balanced between the DPP-4i and TZD groups after 1:3 propensity score matching. The matched cohort (n = 8,468) consisted of 6,351 patients in the DPP-4i group and 2,117 patients in the TZD group. In propensity score-matched Cohort 1, approximately 58% of the patients were in the age group of 45–64 years, and the average BMI was 25.5 kg/m$^2$. (Table 2).

Prior to propensity score matching in Cohort 2, the average age and BMI of SGLT-2i users were higher than those of DPP-4i users. We confirmed that the two groups were well matched following 1:3 propensity score matching, and that the number of patients included in the final analysis was 6,783 and 2,261 in the DPP-4i and SGLT-2i groups, respectively. Approximately 80% of patients in matched cohort 2 had dyslipidemia, with an average BMI of 26.3 kg/m$^2$ and a mean duration of diabetes of 7 years (Table 3).

Of the 8,468 study patients included in cohort 1,998 developed NAFLD during a median follow-up of 1.2 years. The crude incidence of NAFLD according to the total observational person-years was 17,486.6 person-years was 57.1 cases per 1,000 persons per year. The median follow-up period was 1.4 years in the DPP-4i group and 0.8 years in the TZD group. The crude incidence rates were 60.5 cases per 1,000 person-years in the DPP-4i group and 42.9 cases per 1,000 person-years in the TZD group.

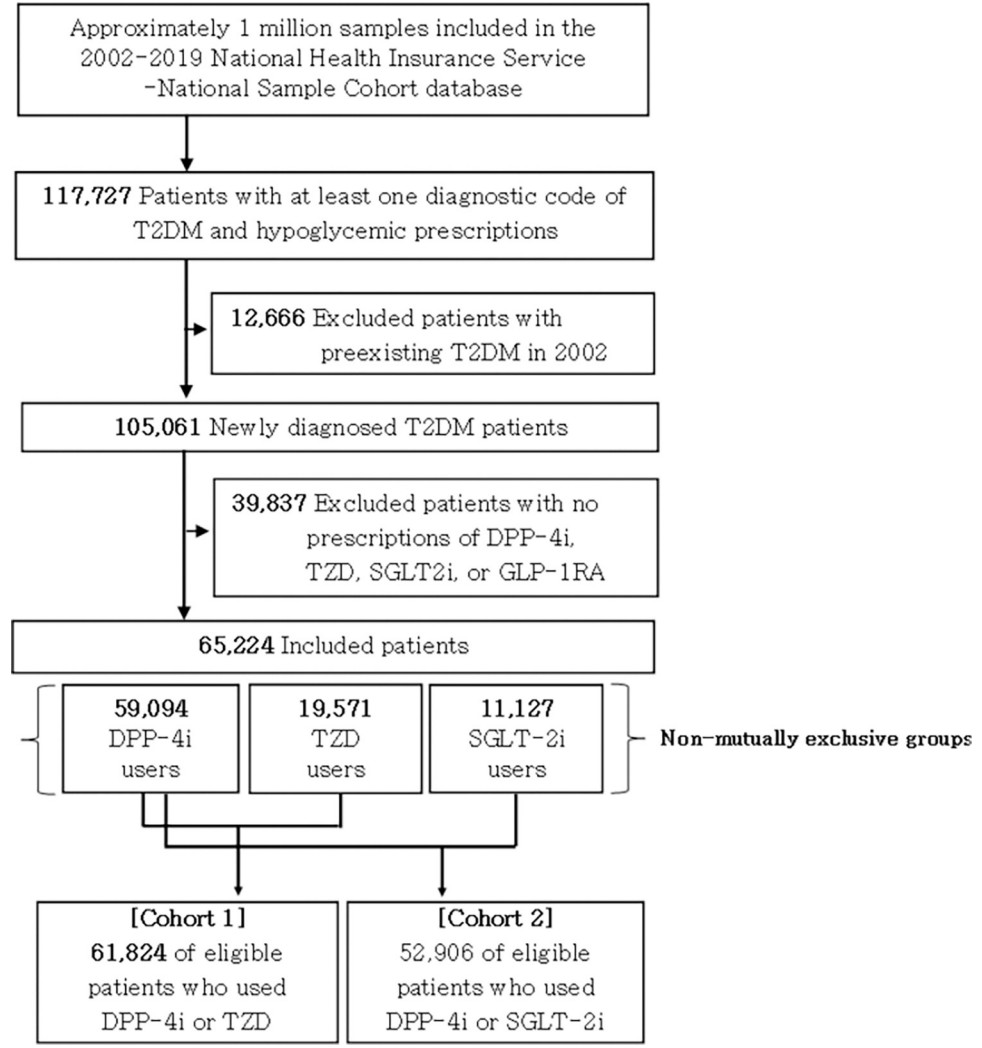

**Fig 1. Flow diagram of inclusion of study participants.**

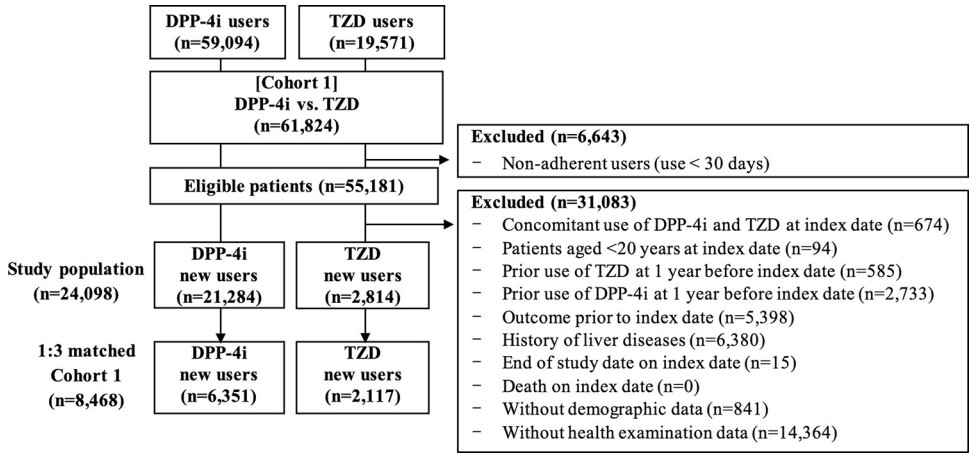

**Fig 2. Flow chart of study selection in Cohort 1.**

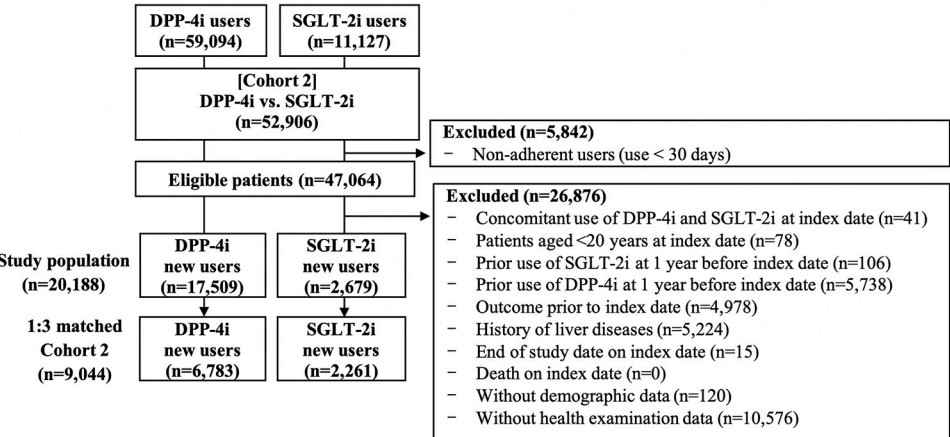

**Fig 3. Flow chart of study selection in Cohort 2.**

TZD use was found to be significantly associated with a 34% NAFLD risk reduction compared to DPP-4i use (HR, 0.66; 95% CI: 0.55–0.78). Of the 998 NAFLD cases, 876 and 122 corresponded to NAFLD (K76.0) and NASH (K75.8), respectively. When outcome was assessed respectively, the HR for TZD use in NASH (K75.8) was 0.37 (95% CI; 0.20–0.69), and 0.69 (95% CI: 0.58–0.83) in NAFLD (K76.0) (Table 4).

Among the 9,044 study participants included in Cohort 2 after 1:3 propensity score matching, 1151 cases of NAFLD occurred during the median follow-up period of 1.2 years. The crude incidence of NAFLD in 15,365.3 person-years was 74.9 cases per 1,000 person-years. The median follow-up period was 1.4 years in the DPP-4i group and 0.8 years in the SGLT-2i group. The crude incidence rates were 74.3 cases per 1,000 person-years in the DPP-4i group and 77.6 cases per 1,000 person-years in the SGLT-2i group.

SGLT-2i use was associated with a reduction in the risk of NAFLD compared to the use of DPP-4i, but the result was not statistically significant (HR, 0.93; 95% CI: 0.80–1.08). Furthermore, 957 and 194 cases out of a total of 1151 NAFLD occurrence were determined to be NAFLD (K76.0) and NASH (K75.8), respectively. When analyzed separately, the HR of NAFLD (K76.0) and NASH (75.8) was 0.84 (95% CI: 0.71–1.00, $p = 0.007$), and 1.33 (95% CI, 0.96–1.84) for SGLT-2i group compared to DPP-4i group (Table 5).

The comparison between DPP-4i and TZD in the subgroup analysis showed that the association between TZD use and a reduced risk of developing NAFLD was particularly significant in the group aged 45–54 years, with a BMI <30 kg/m$^2$, and pioglitazone use. Men and women showed significant associations as observed in the main results. However, the effect size was greater in women, which indicated a larger reduction in the risk of developing NAFLD (Men, HR, 0.73; 95% CI: 0.59–0.91 *vs.* Women, HR, 0.56, 95% CI: 0.42–0.75). Subgroup analysis in Cohort 2 showed a consistent trend with the main result, and although the findings were not statistically significant, the dapagliflozin use group (HR, 0.78; 95% CI, 0.63–0.97) and the group with 23 kg/m$^2 \leq$BMI<25 kg/m$^2$ (HR, 0.66; 95% CI, 0.44–0.99) did show a significant association between SGLT-2i use and reduction in NAFLD risk (S2 Fig).

Sensitivity analysis performed by setting the latency period to 30, 60, 90, and 120 days after drug discontinuation showed no effect modification from the main analysis in both Cohort 1 and Cohort 2 (S3 Table).

**Table 2. Baseline characteristics before and after propensity score matching in Cohort 1.**

| Baseline characteristics | Before matching (n = 24,098) | | | After matching (n = 8,468) | | |
|---|---|---|---|---|---|---|
| | DPP-4i users (n = 21,284) | TZD users (n = 2,814) | SMD | DPP-4i users (n = 6,351) | TZD users (n = 2,117) | SMD |
| Sex | | | | | | |
| Men, n (%) | 12,471 (58.6) | 1,637 (58.2) | -0.009 | 3,702 (58.3) | 1,215 (57.4) | -0.018 |
| Women, n (%) | 8,813 (41.4) | 1,177 (41.8) | | 2,649 (41.7) | 902 (42.6) | |
| Age (years), mean ± SD | 58.6 ± 11.5 | 58.4 ± 11.3 | -0.014 | 58.6 ± 11.4 | 58.6 ± 11.5 | -0.001 |
| Age groups | | | 0.006 | | | 0.027 |
| 19–44 years, n (%) | 2,468 (11.6) | 330 (11.7) | | 736 (11.6) | 247 (11.7) | |
| 45–64 years, n (%) | 12,212 (57.4) | 1,606 (57.1) | | 3,664 (57.7) | 1,195 (56.5) | |
| ≥65 years, n (%) | 6,604 (31.0) | 878 (31.2) | | 1,951 (30.7) | 675 (31.9) | |
| BMI (kg/m$^2$) [a], mean ± SD | 25.5 ± 3.6 | 25.5 ± 3.5 | -0.011 | 25.5 ± 3.5 | 25.5 ± 3.5 | 0.001 |
| Comorbidities | | | | | | |
| Dyslipidemia, n (%) | 15,287 (71.8) | 1,723 (61.2) | 0.004 | 4,252 (67.0) | 1,396 (65.9) | 0.002 |
| Hypertension, n (%) | 11,728 (55.1) | 1,575 (56.0) | 0.004 | 3,524 (55.5) | 1,173 (55.4) | 0.002 |
| OSA, n (%) | 24 (0.11) | 1 (0.04) | 0.004 | 3 (0.05) | 1 (0.05) | 0.002 |
| Hypothyroidism, n (%) | 1,370 (6.4) | 153 (5.4) | 0.004 | 375 (4.4) | 136 (6.4) | 0.002 |
| MI, n (%) | 329 (1.6) | 36 (1.3) | 0.004 | 86 (1.4) | 28 (1.3) | 0.002 |
| IHD, n (%) | 1,840 (8.6) | 209 (7.4) | 0.004 | 500 (7.9) | 167 (7.9) | 0.002 |
| CHF, n (%) | 930 (4.4) | 103 (3.7) | 0.004 | 244 (3.8) | 79 (3.7) | 0.002 |
| CCI, n (%) | | | 0.095 | | | 0.013 |
| 0 | 4,005 (18.8) | 623 (22.1) | | 1,304 (20.5) | 425 (20.1) | |
| 1 | 5,953 (28.0) | 704 (25.0) | | 1,646 (25.9) | 552 (26.1) | |
| 2 | 5,036 (23.7) | 644 (22.9) | | 1,496 (23.6) | 496 (23.4) | |
| ≥3 | 6,290 (29.6) | 843 (30.0) | | 1,905 (30.0) | 644 (30.4) | |
| Comedications | | | | | | |
| Methotrexate, n (%) | 18 (0.08) | 2 (0.07) | 0.082 | 18 (0.08) | 2 (0.07) | -0.011 |
| Corticosteroids, n (%) | 889 (4.2) | 75 (2.7) | 0.082 | 197 (3.1) | 62 (2.9) | -0.011 |
| Valproic acid, n (%) | 65 (0.3) | 8 (0.3) | 0.082 | 22 (0.4) | 7 (0.3) | -0.011 |
| Tamoxifen, n (%) | 5 (0.02) | 2 (0.07) | 0.082 | 4 (0.06) | 0 (0) | -0.011 |
| HRT, n (%) | 86 (0.4) | 8 (0.3) | 0.082 | 4 (0.06) | 0 (0) | -0.011 |
| Metformin, n (%) | 18,474 (86.8) | 1,730 (61.5) | 0.082 | 5,021 (79.1) | 1,693 (80.0) | -0.011 |
| Meglitinides, n (%) | 39 (0.2) | 19 (0.7) | | 13 (0.2) | 10 (0.5) | |
| AGI, n (%) | 160 (0.8) | 89 (3.2) | | 80 (1.3) | 31 (1.5) | |
| Sulfonylureas, n (%) | 4,447 (20.9) | 1,323 (47.0) | 0.082 | 1,874 (29.5) | 637 (30.1) | -0.011 |
| Insulin, n (%) | 1,097 (5.2) | 108 (3.8) | 0.082 | 270 (4.3) | 98 (4.6) | -0.011 |
| SGLT-2i, n (%) | 43 (0.2) | 9 (0.3) | 0.082 | 18 (0.3) | 9 (0.4) | -0.011 |
| Surrogate variables for severity of T2DM | | | | | | |
| Number of hypoglycemic agents, n (%) | | | 0.174 | | | 0.093 |
| 1 | 678 (3.2) | 119 (4.2) | | 303 (4.8) | 119 (5.6) | |
| 2 | 9,235 (43.4) | 1,015 (36.1) | | 2,478 (39.0) | 734 (34.7) | |
| 3 | 7,751 (36.4) | 1,059 (37.6) | | 2,394 (37.7) | 844 (39.9) | |
| ≥4 | 3,620 (17.0) | 621 (22.1) | | 1,176 (18.5) | 420 (19.8) | |
| T2DM complications, n (%) | 7,941 (37.3) | 1,215 (43.2) | 0.004 | 2,517 (39.6) | 868 (41.0) | 0.002 |
| T2DM duration (years), mean ± SD | 5.8 ± 4.6 | 5.1 ± 3.8 | -0.178 | 5.4 ± 4.5 | 5.4 ±4.1 | -0.003 |

Abbreviations: SMD, standardized mean difference; SD, standard deviation; BMI, body mass index; DPP-4i, dipeptidyl peptidase-4 inhibitors; TZD, thiazolidinediones; OSA, obstructive sleep apnea; MI, myocardial infarction; IHD, ischemic heart disease; CHF, chronic heart failure; CCI, Charlson comorbidity index; HRT, hormone replacement therapy; AGI, alpha-glucosidase inhibitors; SGLT-2i, sodium-glucose cotransporter-2 inhibitors; T2DM, diabetes mellitus.

[a] Excluding missing values in BMI (n = 1,981).

**Table 3. Baseline characteristics before and after propensity score matching in Cohort 2.**

| Baseline characteristics | Before matching (n = 20,188) | | | After matching (n = 9,044) | | |
|---|---|---|---|---|---|---|
| | DPP-4i users (n = 17,509) | SGLT-2i users (n = 2,679) | SMD | DPP-4i users (n = 6,783) | SGLT-2i users (n = 2,261) | SMD |
| Sex | | | 0.003 | | | |
| Men, n (%) | 10,450 (59.7) | 1,603 (59.8) | | 4,016 (59.2) | 1,332 (58.9) | -0.008 |
| Women, n (%) | 7,059 (40.3) | 1,076 (40.2) | | 2,767 (40.8) | 929 (41.1) | |
| Age (years), mean ± SD | 59.2 ± 11.7 | 55.5 ± 11.6 | | 56.9 ± 11.4 | 56.7 ± 11.3 | |
| Age groups | | | 0.300 | | | 0.041 |
| 19–44 years, n (%) | 1,939 (11.1) | 479 (17.9) | | 982 (14.5) | 325 (14.4) | |
| 45–64 years, n (%) | 9,762 (55.8) | 1,629 (60.8) | | 4,057 (59.8) | 1,393 (61.6) | |
| ≥65 years, n (%) | 5,808 (33.2) | 571 (21.3) | | 1,744 (25.7) | 543 (24.0) | |
| BMI (kg/m$^2$) [a], mean ± SD | 25.5 ± 3.6 | 27.0 ± 4.1 | 0.376 | 26.3 ± 3.6 | 26.3 ± 3.6 | 0.025 |
| Comorbidities | | | | | | |
| Dyslipidemia, n (%) | 12,976 (74.1) | 2,202 (82.2) | 0.194 | 5,403 (79.7) | 1,810 (80.1) | -0.003 |
| Hypertension, n (%) | 9,635 (55.0) | 1,566 (58.5) | 0.194 | 3,847 (56.7) | 1,283 (56.7) | -0.003 |
| OSA, n (%) | 19 (0.1) | 4 (0.2) | 0.194 | 9 (0.1) | 4 (0.2) | -0.003 |
| Hypothyroidism, n (%) | 1,109 (6.3) | 194 (7.2) | 0.194 | 445 (6.6) | 156 (6.9) | -0.003 |
| MI, n (%) | 293 (1.7) | 81 (3.0) | 0.194 | 137 (2.0) | 51 (2.3) | -0.003 |
| IHD, n (%) | 1,484 (8.5) | 355 (13.3) | 0.194 | 729 (10.8) | 238 (10.5) | -0.003 |
| CHF, n (%) | 805 (4.6) | 172 (6.4) | 0.194 | 343 (5.1) | 123 (5.4) | -0.003 |
| CCI, n (%) | | | 0.089 | | | 0.019 |
| 0 | 3,332 (19.0) | 429 (16.0) | | 1,148 (16.9) | 389 (17.2) | |
| 1 | 4,912 (28.1) | 739 (27.6) | | 1,934 (28.5) | 629 (27.8) | |
| 2 | 4,145 (23.7) | 655 (24.5) | | 1,628 (24.0) | 555 (24.6) | |
| ≥3 | 5,120 (29.2) | 856 (32.0) | | 2,073 (30.6) | 688 (30.4) | |
| Comedications | | | | | | |
| Methotrexate, n (%) | 15 (0.1) | 1 (0.0) | -0.079 | 0 (0) | 1 (0.04) | 0.007 |
| Corticosteroids, n (%) | 745 (4.3) | 91 (3.4) | -0.079 | 249 (3.7) | 83 (3.7) | 0.007 |
| Valproic acid, n (%) | 51 (0.3) | 5 (0.2) | -0.079 | 9 (0.1) | 5 (0.2) | 0.007 |
| Tamoxifen, n (%) | 4 (0.0) | 0 (0) | -0.079 | 0 (0) | 0 (0) | 0.007 |
| HRT, n (%) | 60 (0.3) | 14 (0.5) | -0.079 | 30 (0.4) | 9 (0.4) | 0.007 |
| Metformin, n (%) | 14,828 (84.7) | 2,202 (82.2) | -0.079 | 5,646 (83.2) | 1,882 (83.2) | 0.007 |
| Meglitinides, n (%) | 21 (0.1) | 0 (0) | -0.079 | 0 (0) | 0 (0) | 0.007 |
| AGI, n (%) | 81 (0.5) | 8 (0.3) | -0.079 | 18 (0.3) | 8 (0.4) | 0.007 |
| Sulfonylureas, n (%) | 3,799 (21.7) | 736 (27.5) | -0.079 | 1,672 (24.7) | 555 (24.6) | 0.007 |
| Insulin, n (%) | 1,079 (6.2) | 142 (5.3) | -0.079 | 380 (5.6) | 123 (5.4) | 0.007 |
| TZD, n (%) | 612 (3.5) | 32 (1.2) | -0.079 | 84 (1.2) | 32 (1.4) | 0.007 |
| Surrogate variables for severity of T2DM | | | | | | |
| Number of hypoglycemic agents, n (%) | | | 0.195 | | | 0.038 |
| 1 | 631 (3.6) | 151 (5.6) | | 330 (4.9) | 94 (4.2) | |
| 2 | 7,694 (43.9) | 1,045 (39.0) | | 2,859 (42.2) | 952 (42.1) | |
| 3 | 6,319 (36.1) | 879 (32.8) | | 2,308 (34.0) | 792 (35.0) | |
| ≥4 | 2,865 (16.4) | 604 (22.6) | | 1,286 (19.0) | 423 (18.7) | |
| T2DM complications, n (%) | 6,369 (36.4) | 1,099 (41.0) | 0.194 | 2,690 (39.7) | 877 (38.8) | -0.003 |
| T2DM duration (years), mean ± SD | 6.5 ± 5.0 | 7.0 ± 5.2 | 0.096 | 6.9 ± 5.0 | 6.7 ± 5.2 | -0.031 |

Abbreviations: SMD, standardized mean difference; SD, standard deviation; BMI, body mass index; DPP-4i, dipeptidyl peptidase-4 inhibitors;; SGLT-2i, sodium-glucose cotransporter-2 inhibitors; OSA, obstructive sleep apnea; MI, myocardial infarction; IHD, ischemic heart disease; CHF, chronic heart failure; CCI, Charlson comorbidity index; HRT, hormone replacement therapy; AGI, alpha-glucosidase inhibitors; TZD, thiazolidinediones; T2DM, type 2 diabetes mellitus.

[a] Excluding missing values in BMI (n = 2,226).

**Table 4. Incidence rate and the risk of nonalcoholic fatty liver disease in Cohort 1.**

| Cohort | Exposure | No. of patients | Follow-up years, median (IQR) | Total person-year | No. of events | Crude IR | PS-matched HR (95% CI) |
|---|---|---|---|---|---|---|---|
| [Cohort 1] DPP-4i *vs.* TZD (n = 8,468) | NAFLD[a] | | | | | | |
| | DPP-4i | 6,351 | 1.4 (0.4–3.4) | 14,061.5 | 851 | 60.5 | 1.00 (ref) |
| | TZD | 2,117 | 0.8 (0.2–2.2) | 3,425 | 147 | 42.9 | 0.66 (0.55–0.78)[b] |
| | Total | 8,468 | 1.2 (0.3–3.1) | 17,486.6 | 998 | 57.1 | |
| | NAFLD (K76.0) | | | | | | |
| | DPP-4i | 6,240 | 1.4 (0.4–3.4) | 13,882.9 | 740 | 53.3 | 1.00 (ref) |
| | TZD | 2,106 | 0.8 (0.2–2.2) | 3,403.1 | 136 | 40.0 | 0.69 (0.58–0.83) |
| | Total | 8,346 | 1.2 (0.4–3.1) | 17,286 | 876 | 50.7 | |
| | NASH (K75.8) | | | | | | |
| | DPP-4i | 5,611 | 1.5 (0.4–3.6) | 12,883 | 111 | 8.62 | 1.00 (ref) |
| | TZD | 1,981 | 0.8 (0.3–2.2) | 3,255.6 | 11 | 3.38 | 0.37 (0.20–0.69) |
| | Total | 7,592 | 1.3 (0.4–3.2) | 16,138.6 | 122 | 7.56 | |

Abbreviations: IQR, interquartile range; IR, incidence rate; PS, propensity score; HR, hazard ratio; CI, confidence interval; DPP-4i, dipeptidyl peptidase-4 inhibitors; TZD, thiazolidinediones; NAFLD, nonalcoholic fatty liver disease; NASH, nonalcoholic steatohepatitis.

[a] Refers to outcome including both NAFLD (K76.0) and NASH (K75.8).

## Discussion

This study aimed to assess the impact of a particular classes of hypoglycemic agents on the risk of NAFLD in patients with type 2 diabetes using large-scale cohort data from the NHIS spanning from 2002 to 2019. Our results indicate that the use of TZD was associated with a significantly lower risk of NAFLD compared to the use of DPP-4i. A similar trend was observed when comparing SGLT-2i and DPP-4i, but this association was not statistically significant.

Of 65,224 patients newly diagnosed with type 2 diabetes between 2003 and 2019, 42.6% had a diagnostic history of NAFLD. When analyzed according to BMI group, the prevalence of

**Table 5. Incidence rate and the risk of nonalcoholic fatty liver disease in Cohort 2.**

| Cohort | Exposure | No. of patients | Follow-up years, median (IQR) | Total person-year | No. of events | Crude IR | PS-matched HR (95% CI) |
|---|---|---|---|---|---|---|---|
| [Cohort 2] DPP-4i *vs.* SGLT-2i (n = 9,044) | NAFLD[a] | | | | | | |
| | DPP-4i | 6,783 | 1.4 (0.4–3.0) | 12,567.4 | 934 | 74.3 | 1.00 (ref) |
| | SGLT-2i | 2,261 | 0.8 (0.3–1.9) | 2,797.8 | 217 | 77.6 | 0.93 (0.80–1.08) |
| | Total | 9,044 | 1.2 (0.4–2.7) | 15,365.3 | 1151 | 74.9 | |
| | NAFLD (K76.0) | | | | | | |
| | DPP-4i | 6,638 | 1.4 (0.4–3.0) | 12,361.5 | 789 | 63.8 | 1.00 (ref) |
| | SGLT-2i | 2,212 | 1.5 (0.5–3.1) | 2,766.7 | 168 | 60.7 | 0.84 (0.71–1.00)[b] |
| | Total | 8,850 | 1.2 (0.4–2.7) | 15,128.3 | 957 | 63.3 | |
| | NASH (K75.8) | | | | | | |
| | DPP-4i | 5,994 | 1.5 (0.5–3.1) | 1,1428.7 | 145 | 12.7 | 1.00 (ref) |
| | SGLT-2i | 2,093 | 0.9 (0.3–2.0) | 2,459.5 | 49 | 19.9 | 1.33 (0.96–1.84) |
| | Total | 8,087 | 1.3 (0.4–2.8) | 14,252.3 | 194 | 13.6 | |

Abbreviations: IQR, interquartile range; IR, incidence rate; PS, propensity score; HR, hazard ratio; CI, confidence interval; DPP-4i, dipeptidyl peptidase-4 inhibitor; TZD, thiazolidinedione; NAFLD, nonalcoholic fatty liver disease; NASH, nonalcoholic steatohepatitis.

[a] Refers to outcome including both NAFLD (K76.0) and NASH (K75.8).

[b] $p = 0.007$.

NAFLD increased significantly with higher BMI: a prevalence of 54.5% was observed in patients with BMI of $\geq$30 kg/m$^2$, and the same trend was also reported in previous studies [18]. Additionally, previous studies have reported varied prevalence of NAFLD among patients with type 2 diabetes, depending on the diagnostic method. A study that identified NAFLD through abdominal ultrasonography reported a prevalence of 27.3% [19]. The differences between these results can be partly attributed to the characteristics of the claims data, which did not provide values of specific test results or clinical findings.

TZD is a hypoglycemic agent that improves insulin sensitivity in the adipose tissue, liver, and muscle. Numerous clinical studies have shown that pioglitazone improves hepatocellular damage and hepatic fibrosis in patients with histologically confirmed NASH [20,21]. These clinical effects may suppress the differentiation of adipocytes, prevent the accumulation of excess triglycerides in the liver, and reduce inflammation in the liver by lowering the levels of inflammatory factors such as adiponectin or highly sensitive C-reactive protein [22]. Several retrospective studies have suggested that TZD may not only offer liver benefits, but also help prevent NAFLD. A retrospective cohort study that utilized UK claims data found a significant association between the use of TZD and a 68% reduction in the risk of developing NAFLD [13]. The finding from this study is in accordance with the results of the present study, in which we confirmed a 34% reduction in the risk of NAFLD associated with TZD use. The median follow-up durations were 1.4 and 0.8 years for DPP-4i and TZD, respectively, and this discrepancy in the follow-up period was also found in previous studies. It can be explained by relatively better tolerance of DPP-4i compared to TZD, which may have resulted in the higher persistency in DPP-4i group than in TZD group [23].

While no effect modification was found in comparison to the main analysis in the subgroup analysis of Cohort 1, TZD use and the reduced risk of NAFLD was statistically significant especially in those who used pioglitazone or in those aged 45–65 years. In addition, the association was not significant in the group with a BMI of >30 kg/m$^2$. This may be attributed to the fact that as obesity is major risk factor for NAFLD, the beneficial effect of TZD on the NAFLD may have been negated by the effect of underlying obesity. However, the small sample size in the group with a BMI of >30 kg/m$^2$ compared to the other BMI groups cannot be ignored, as it may resulted in insufficient power. Sensitivity analysis according to the various latency periods and drug exposure yielded consistent trends and statistical significance confirming the robustness of the results.

Overall, TZD may be used to reduce the risk of NAFLD in patients with type 2 diabetes. This finding supports the clinical implications of choosing TZD over DPP-4i in patients with type 2 diabetes in terms of reducing the risk of NAFLD. However, it is important to consider potential risks of using TZD, such as the risk of weight gain or heart failure. This necessitates a weighing of risks versus benefits for TZD use on an individual patient basis, and patients should be closely monitored for adverse effects [24,25].

On the other hand, SGLT-2i lowers blood sugar by inhibiting the receptors involved in glucose reabsorption in the proximal tubules to promote glucose excretion [26]. In addition to its hypoglycemic effect, SGLT-2i have been shown to provide significant benefits for weight loss and cardiovascular health [27,28]. Studies have shown that weight loss can improve insulin sensitivity and reduce liver fat, which can ultimately prevent NAFLD or improve existing liver diseases [29]. One randomized clinical trial demonstrated that 20 weeks of empagliflozin treatment significantly improved fatty liver and liver enzyme levels in patients with type 2 diabetes compared to those in the control group [30]. In addition, dapagliflozin use for 12 weeks in patients with type 2 diabetes and NAFLD lowered factors related to hepatocellular damage and improved fatty liver [31].

In the main analysis of Cohort 2, the use of SGLT-2i was not significantly associated with the reduction in the risk of NAFLD compared with DPP-4i group. Median follow-up

durations were longer in DPP-4i compared to that of SGLT-2i. Similar to the discrepancy in the follow-up period in Cohort 1, this is speculated as the result that reflects poor adherence of SGLT-2i [32]. When analyzed by subgroups within Cohort 2, the results were mostly in line with the main analysis except in those who used ipragliflozin (HR, 1.21; 95% CI: 0.63–2.33) and in those with BMI $<23$ kg/m$^2$ (HR, 1.11; 95% CI: 0.44–0.99), but with no statistical significance. Relatively small number of patients who used ipragliflozin compared to the other SGLT-2i drugs may be attributed to this result (e.g., ipragliflozin users (n = 88) vs. dapagliflozin users (n = 1075)), or from a distinct phenotype of NAFLD in those with BMI $<23$ kg/m$^2$, who are referred as 'lean NAFLD', since these patients are known to have lower metabolic risk but higher fibrosis score and cardiovascular risk than NAFLD within other BMI groups [33]. The findings of the current study are consistent with those of previous retrospective studies, which have also suggested the potential benefits of SGLT-2i in reducing the risk of NAFLD. A retrospective study verified that the use of SGLT-2i in patients with type 2 diabetes was associated with a 22% reduction in the risk of developing NAFLD compared with the use of DPP-4i [12]. In addition, a study using real-world data in South Korea showed a significant improvement in the fatty liver index—a prediction score for NAFLD—and compared SGLT-2i with DPP-4i [34]. Overall, although the estimated effect has not shown statistical significance, the results of this study corroborates previous findings that SGLT-2i could be preferentially considered over DPP-4i for lowering a potential risk of NAFLD in patients with type 2 diabetes.

To the best of our knowledge, this is the first study to examine the association between the risk of NAFLD occurrence and the use of hypoglycemic drugs in patients with type 2 diabetes in South Korea population. The results of this study gained more power and generalizability by utilizing a nationwide database that captured real-world clinical practice. Also, through an active comparator and new user cohort design and propensity score matching, the independent effect of hypoglycemic drug use on the risk of NAFLD was evaluated by controlling for covariates, including demographic characteristics, comorbidities, and medications, between the comparator and treatment groups. Furthermore, we demonstrated robustness of the results through sensitivity analysis based on varying latency period.

Despite these strengths, this study has several limitations. Firstly, owing to the limited clinical information in NHIS database, we used ICD-10 codes to define the occurrence of NAFLD, which may have led to underestimation of the outcome [35]. Although defining NAFLD using disease codes is considered as an alternative option when data source lacks clinical information given the reported accuracy of 85% for defining NAFLD, its accuracy within the Korean population has not been fully validated [36,37]. Therefore, further studies that incorporate clinical variables such as NAFLD prediction index or imaging data are needed to validate the current findings.

Additionally, the follow-up period in this study may not have been sufficient to capture the long-term effect of hypoglycemic agents [38,39]. Limited follow-up period may be a result of as-treated analysis approach, where the discontinuation or add-on of other hypoglycemic agents was considered as a censoring event. However, this approach was necessary to reflect a real-world treatment pattern of hypoglycemic agents, where switching and add-on happens commonly [40]. Lastly, the findings of this study should be interpreted with caution considering the possible bias despite the application of various statistical measures. For instance, including BMI or comedication information that have been captured during the follow up period may have introduced post-treatment bias, and excluding non-adherent patients by assessing 30 days after the index date may potentially led to selection bias. Thus, while this study offers valuable insights into NAFLD management in patients with type 2 diabetes, further research that validate these findings through rigorous statistical approach are needed.

In conclusion, the findings of this study add to the growing body of evidence supporting the potential benefits of TZD and SGLT-2i in lowering the risk of NAFLD in patients with

type 2 diabetes mellitus. We expect the potential benefit of TZD, and possibly SGLT-2i, for reducing the risk of NAFLD occurrence can support the development of effective pharmacotherapy strategies for preventing and managing NAFLD in patients with type 2 diabetes mellitus. However, prospective studies and studies that use more detailed diagnostic information are necessary to support these findings.

## Supporting information

**S1 Checklist. STROBE statement—Checklist of items that should be included in reports of observational studies.**
(DOCX)

**S1 Fig. Graphical depiction of study design.**
(TIF)

**S2 Fig.** Subgroup analysis according to age, sex, body mass index, and ingredients (a) Cohort 1 (b) Cohort 2.
(TIF)

**S1 Table. Diagnostic codes used to identify diseases of interest.**
(DOCX)

**S2 Table. Prevalence of nonalcoholic fatty liver disease among patients with type 2 diabetes.**
(DOCX)

**S3 Table. Sensitivity analysis based on latency periods.**
(DOCX)

## Author Contributions

**Conceptualization:** Hyunah Kim.

**Formal analysis:** Choungwon Jung, Soyoung Park.

**Funding acquisition:** Hyunah Kim.

**Investigation:** Choungwon Jung, Soyoung Park, Hyunah Kim.

**Methodology:** Choungwon Jung, Soyoung Park, Hyunah Kim.

**Project administration:** Hyunah Kim.

**Resources:** Hyunah Kim.

**Supervision:** Hyunah Kim.

**Validation:** Hyunah Kim.

**Visualization:** Choungwon Jung, Soyoung Park.

**Writing – original draft:** Choungwon Jung, Soyoung Park.

**Writing – review & editing:** Hyunah Kim.

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
