## [Decision Letter · Decision Letter 0]

6 Jun 2023

PONE-D-23-12492Association between hypoglycemic agent use and the risk of occurrence of nonalcoholic fatty liver disease in patients with type 2 diabetes mellitusPLOS ONE

Dear Dr. Kim,

Thank you for submitting your manuscript to PLOS ONE. After careful consideration, we feel that it has merit but does not fully meet PLOS ONE’s publication criteria as it currently stands. Therefore, we invite you to submit a revised version of the manuscript that addresses the points raised during the review process.

We look forward to receiving your revised manuscript.

Kind regards,

Han Eol Jeong, M.P.H., Ph.D.

Academic Editor

PLOS ONE

“This work was supported by the National Research Foundation of Korea (NRF) grant funded by the Korean government (MSIT) [grant number NRF-2021R1F1A1052976].”

Reviewers' comments:

Reviewer's Responses to Questions

**Comments to the Author**

1. Is the manuscript technically sound, and do the data support the conclusions?

Reviewer #1: Partly

Reviewer #2: Partly

2. Has the statistical analysis been performed appropriately and rigorously? 

Reviewer #1: No

Reviewer #2: No

3. Have the authors made all data underlying the findings in their manuscript fully available?

Reviewer #1: Yes

Reviewer #2: Yes

4. Is the manuscript presented in an intelligible fashion and written in standard English?

Reviewer #1: Yes

Reviewer #2: No

5. Review Comments to the Author

Reviewer #1: 1. Propensity score estimation

There should be more clarity on how propensity scores were calculated.

<1> page 8, line 175-176: “The propensity score of individuals was calculated by considering covariates through a multivariate logistic regression analysis.”

<2> page 8, line 167-171: “The BMI of each patient was assessed for a period of 1 year prior to and 1 year after the index date…”, “Concomitant medications were defined as medications that individuals take for at least one overlapping day within the follow-up period.”

-> Do the covariates mentioned in the sentence in <1> include BMI and concomitant medications? If so, based on the statements in <2>, it is concerning that post-index date covariates were included in the model used to calculate the propensity scores.

This raises concerns because including post-baseline covariates may introduce biases influenced by treatment. Especially considering that both TZD and SGLT2 inhibitors, which are the drugs of interest in this study, are known to affect body weight when taken. Therefore, selecting post-treatment BMI as a conditional covariate for calculating the propensity score (the probability of treatment assignment) is not reasonable. Thus, it is necessary to specify the rationale for selecting covariates in propensity score calculation and describe that the propensity scores were calculated using an appropriate model.

2. Outcome definition

It is highly likely that the variability in the prevalence of NAFLD across studies, as mentioned by the authors on “page 16, lines 310-317”, is due to the lack of a valid definition for NAFLD. In this study, NAFLD was defined as K76.0 and K75.8.

-> Has there been a validation study conducted to define NAFLD using diagnostic codes in the NHIS sample cohort? If such a study exists, it would be advisable to mention it. There is a need to strengthen the description of the validity of the outcome. Alternatively, sensitivity analysis should be performed to validate the robustness of the outcome definition.

3. Sensitivity analysis

There are some unclear aspects regarding the sensitivity analysis that applied alternative criteria to reassign the treatment groups. It is necessary to provide clearer descriptions in this regard.

<1> Page 9, line 190-192: “…, alternative criterion assigned patients to groups based on the drug they used for the longest period during the follow-up period.”

<2> Page 18, line 361-363: “Interestingly, the proportion of SGLT-2i users in Cohort 2 increased slightly from 13.3% in main analysis to 14.6% in sensitivity analysis, which indicates that some patients may have switched from DPP-4i to SGLT-2i and used SGLT-2 inhibitors for longer periods than DPP-4i.”

According to the statements in <1> and <2>, if a “patient A” initially used DPP-4i and later switched to SGLT-2i, and the duration of SGLT-2i use was longer than the duration of DPP-4i use, then this patient would be considered as part of the SGLT-2i group when calculating the hazard ratio (HR).

-> If “patient A” switches from DPP4i to SGLT2i, is the person-time calculated using the new index date as the date “patient A” starts SGLT2i? Then, what about the person-time prior to the switch, i.e. while taking DPP4i? Did you include that in the person-time for the DPP4i group or ignore it?

-> If “patient A” experience an outcome while on DPP4i, is this patient censored, or will he/she be assigned to the SGLT2i group (in the future) because they have been on SGLT2i for longer, so the outcome before they are assigned to SGLT2i is not considered?

Reviewer #2: This retrospective cohort study examined the risk of nonalcoholic fatty liver disease (NAFLD) in patients with type 2 diabetes using different hypoglycemic agents. The study utilized data from the National Health Insurance Service–National Sample Cohort of South Korea. The findings showed that the use of thiazolidinediones (TZD) was associated with a reduced risk of NAFLD compared to dipeptidyl peptidase-4 inhibitors (DPP-4i). Sodium-glucose cotransporter-2 inhibitors (SGLT-2i) also demonstrated a lower risk of NAFLD, although not statistically significant. Sensitivity analysis considering the longer duration of drug use further supported the protective effect of SGLT-2i on NAFLD. These results suggest that prioritizing TZD and SGLT-2i over DPP-4i may help decrease the risk of NAFLD in patients with type 2 diabetes.

However, there are several major points that need to be addressed. In particular, the method for determining the NAFLD outcome needs to be revised.

1. Due to the low diagnostic validity of NAFLD, it is not possible to capture the outcome using ICD codes. To determine the outcome, validity data for the outcome or algorithms such as those described in the following reference should be applied: Pradhan R, Yin H, Yu O, Azoulay L. Glucagon-Like Peptide 1 Receptor Agonists and Sodium-Glucose Cotransporter 2 Inhibitors and Risk of Nonalcoholic Fatty Liver Disease Among Patients With Type 2 Diabetes. Diabetes Care. 2022 Apr 1;45(4):819-829. doi: 10.2337/dc21-1953. PMID: 35104330.

2. If exclusions are made after the follow-up period following propensity score matching, the balance achieved by the propensity score matching may be compromised. To incorporate adherence, it is recommended to utilize indicators such as PDC, MPR, and perform subgroup analyses.

3. There are no references provided for key claims (e.g., the neutral effect of DPP-4i on liver fat or the definition of the outcome).

4. The exposure assessment method is not clearly described. It seems to have adopted an as-treated approach, but the description needs to be clarified.

5. The median follow-up for TZD and SGLT2i is significantly lower than that for DPP-4i in relation to the outcome of NASH. The reason for this discrepancy should be addressed.

6. The point estimate of SGLT2i shows different trends in certain subgroups. The reason for this discrepancy should be discussed.

7. The flow chart in Figure 1 may be misleading to readers, especially regarding the process of constructing cohort 1 and cohort 2. Comments explaining this process should be provided.

8. It is recommended to conduct sensitivity analyses in various ways. This includes not only intention-to-treat analysis but also exploring other PS techniques such as outcome definition modification or IPTW to measure average treatment effects among the whole population.

9. Line 386: NASH (75.8) should be corrected to NASH (K75.8).

6. PLOS authors have the option to publish the peer review history of their article (what does this mean?). If published, this will include your full peer review and any attached files.

Reviewer #1: No

Reviewer #2: No

---

## [Author Response · Author response to Decision Letter 0]

21 Jul 2023

Thank you for thoughtful suggestions and valuable insights. Point-by-point responses to the review's comments have been uploaded as a separate file.

---

## [Decision Letter · Decision Letter 1]

27 Jul 2023

PONE-D-23-12492R1Association between hypoglycemic agent use and the risk of occurrence of nonalcoholic fatty liver disease in patients with type 2 diabetes mellitusPLOS ONE

Dear Dr. Kim,

Thank you for submitting your manuscript to PLOS ONE. After careful consideration, we feel that it has merit but does not fully meet PLOS ONE’s publication criteria as it currently stands. Therefore, we invite you to submit a revised version of the manuscript that addresses the points raised during the review process.

We look forward to receiving your revised manuscript.

Kind regards,

Han Eol Jeong, M.P.H., Ph.D.

Academic Editor

PLOS ONE

Reviewers' comments:

Reviewer's Responses to Questions

**Comments to the Author**

1. If the authors have adequately addressed your comments raised in a previous round of review and you feel that this manuscript is now acceptable for publication, you may indicate that here to bypass the “Comments to the Author” section, enter your conflict of interest statement in the “Confidential to Editor” section, and submit your "Accept" recommendation.

Reviewer #1: (No Response)

Reviewer #2: (No Response)

2. Is the manuscript technically sound, and do the data support the conclusions?

Reviewer #1: Partly

Reviewer #2: Partly

3. Has the statistical analysis been performed appropriately and rigorously? 

Reviewer #1: No

Reviewer #2: No

4. Have the authors made all data underlying the findings in their manuscript fully available?

Reviewer #1: No

Reviewer #2: No

5. Is the manuscript presented in an intelligible fashion and written in standard English?

Reviewer #1: Yes

Reviewer #2: Yes

6. Review Comments to the Author

Reviewer #1: If patients who switched from DPP4i to SGLT2i were not taken into account for person-time during their DPP4i usage period when calculating the hazard ratio, there is a possibility that the relative risk derived in this manner may overestimate the preventive effects of SGLT2i.

Furthermore, if individuals who experienced the outcome while using other medications before the new-index date were excluded, it implies that susceptible individuals were depleted, leading to potential selection bias.

Considering that this sensitivity analysis may cause confusion among readers, I think it’s better to leave it out.

Reviewer #2: 1. To my knowledge NHIS-NSC data source they used provides the same format of data within 5 months upon request. Additionally, it is known to support storage for further analysis. I recommend authors to conduct additional analysis

2. The current sensitivity analysis appears to be an analysis of prevalent users. Please clarify and describe what is meant by "prevalent user" in this context.

3. Please perform intention-to-treat analysis for the sensitivity analysis.

4. I have already suggested ways to improve outcome validity (e.g., diabetes care). Or authors might be possible to utilize the Fatty Liver Index or Hepatic Steatosis Index. It should be noted that using ICD-10 for analysis may lead to significant underestimation of NAFLD and NASH and has not been validated in the Korean population. Therefore, addressing this issue is essential.

5. Please provide a description of how missing values were handled in the BMI data.

7. PLOS authors have the option to publish the peer review history of their article (what does this mean?). If published, this will include your full peer review and any attached files.

Reviewer #1: No

Reviewer #2: No

---

## [Author Response · Author response to Decision Letter 1]

29 Oct 2023

We have uploaded the rebuttal letter as a separate file labeled 'Response to Reviewers'.

---

## [Editor Report · Decision Letter 2]

2 Nov 2023

Association between hypoglycemic agent use and the risk of occurrence of nonalcoholic fatty liver disease in patients with type 2 diabetes mellitus

PONE-D-23-12492R2

Dear Dr. Kim,

We’re pleased to inform you that your manuscript has been judged scientifically suitable for publication and will be formally accepted for publication once it meets all outstanding technical requirements.

Kind regards,

Han Eol Jeong, M.P.H., Ph.D.

Academic Editor

PLOS ONE

Additional Editor Comments (optional):

Reviewers and myself are both content with the authors' response.
---

## [Editor Report · Acceptance letter]

13 Nov 2023

PONE-D-23-12492R2 

Association between hypoglycemic agent use and the risk of occurrence of nonalcoholic fatty liver disease in patients with type 2 diabetes mellitus 

Dear Dr. Kim:

I'm pleased to inform you that your manuscript has been deemed suitable for publication in PLOS ONE. Congratulations! Your manuscript is now with our production department. 

Kind regards, 

on behalf of

Dr. Han Eol Jeong 

Academic Editor

PLOS ONE